# Glycemic fluctuations, fatigue, and sleep disturbances in type 2 diabetes during ramadan fasting: A cross-sectional study

Satwika Arya Pratama[1☯], Rudy Kurniawan[2☯], Hsiao-Yean Chiu[3,4,5,6], Hsuan-Ju Kuo[7], Emmanuel Ekpor[8], Po-Jen Kung[9,10], Safiruddin Al Baqi[11], Faizul Hasan[12], Debby Syahru Romadlon[12]*

1 Department of Nutrition, Faculty of Sport and Health Sciences, Universitas Negeri Surabaya, Surabaya, Indonesia, 2 Diabetes Connection Care, Eka Hospital Bumi Serpong Damai, Tangerang, Indonesia, 3 School of Nursing, College of Nursing, Taipei Medical University, Taipei, Taiwan, 4 Research Center of Sleep Medicine, College of Medicine, Taipei Medical University, Taipei, Taiwan, 5 Department of Nursing, Taipei Medical University Hospital, Taipei, Taiwan, 6 Research Center of Sleep Medicine, Taipei Medical University Hospital, Taipei, Taiwan, 7 School of Nursing, College of Medicine, National Taiwan University, Taipei, Taiwan, 8 School of Nursing, University of Ghana, Legon, Ghana, 9 Department of Nursing, College of Medicine, National Cheng Kung University, Tainan, Taiwan, 10 School of Nursing, Johns Hopkins University, Baltimore, Maryland, United States, 11 Faculty of Education and Teaching Sciences, State Islamic Institute of Ponorogo, Ponorogo, Indonesia 12 Faculty of Nursing, Chulalongkorn University, Bangkok, Thailand,

☯ Satwika Arya Pratama and Rudy Kurniawan contributed equally to this work as the first authors
* debbysyahru.r@chula.ac.th

## Abstract

### Background

This study aimed to assess the prevalence of glycemic fluctuations, fatigue, and sleep disturbances during Ramadan, and to identify factors associated with hypoglycemia and hyperglycemia events in this period.

### Methods

A cross-sectional study of 88 individuals with type 2 diabetes during Ramadan fasting from (08/03/2024) until (20/04/2024) was conducted. HbA1c levels before Ramadan were obtained from medical records. Participants monitored blood glucose twice daily (during the day and two hours after breaking fast). Blood glucose under 70 mg/dl was considered hypoglycemia, and over 200 mg/dl was hyperglycemia. Fatigue was thoroughly assessed using the Indonesian Multidimensional Fatigue Inventory-20 (IMFI-20), while the Pittsburgh Sleep Quality Index (PSQI) was utilized to evaluate sleep quality. In addition, data on sleep duration, as well as dietary habits during Ramadan, were also collected.

### Results

A total of 88 patients with type 2 diabetes (mean age, 52.7 years) participated, predominantly female (68.2%) and married (63.6%). The study found a prevalence of 21.6% for hypoglycemia and 30.6% for hyperglycemia. Additionally, 30.7% of participants experienced fatigue, and 40.9% reported poor sleep quality. HbA1c levels before Ramadan

**Data availability statement:** All relevant data are within the paper and its Supporting Information files.

**Funding:** The author(s) received no specific funding for this work.

**Competing interests:** The authors declare no conflicts of interest.

and fatigue were significantly associated with both hypoglycemia and hyperglycemia (p < 0.05). Sleep quality was also significantly associated with hyperglycemia events (p < 0.05). Furthermore, sleep duration was significantly related to hyperglycemia events (p = 0.01). Meal timing, frequency, and dietary patterns during Ramadan were also found to be significantly associated with both hypoglycemia and hyperglycemia (both p < 0.05).

## Conclusion

Hypoglycemia and hyperglycemia are common among people with type 2 diabetes during Ramadan. Fatigue and poor sleep quality were also widespread. Key factors linked to these glycemic fluctuations were pre-Ramadan HbA1c levels and fatigue, while sleep quality was particularly associated with hyperglycemia. These results highlight the need for personalized care to manage blood sugar levels and improve overall health during Ramadan. We recommend that healthcare providers advise patients with type 2 diabetes to aim for 7–8 hours of sleep per night to help control blood glucose levels. Additionally, having three meals a day (Suhoor, Iftar, and a post-Iftar snack) with low glycemic index foods can help maintain stable blood glucose and prevent both hypoglycemia and hyperglycemia during Ramadan.

## 1 Introduction

Type 2 diabetes mellitus is a prevalent chronic metabolic disorder characterized by insulin resistance and relative insulin deficiency, leading to hyperglycemia [1]. It is associated with various complications, including cardiovascular diseases, neuropathy, retinopathy, and nephropathy, which significantly impact the quality of life and overall health outcomes of affected individuals [2]. The management of type 2 diabetes typically involves lifestyle modifications, pharmacotherapy, and continuous monitoring of blood glucose levels to mitigate the risks of complications. Among the lifestyle modifications, dietary habits play a crucial role in maintaining glycemic control [3]. Ramadan, the holy month of fasting observed by Muslims worldwide, poses unique challenges for individuals with type 2 diabetes due to the significant alterations in meal patterns and fasting durations.

Fasting during Ramadan involves abstaining from food and drink from dawn until sunset, which can lead to substantial changes in blood glucose levels [4]. These fluctuations in glycemia can exacerbate symptoms of fatigue and disturb sleep patterns in individuals with type 2 diabetes [5]. Adequate sleep is crucial for maintaining blood glucose regulation in individuals with diabetes [6]. Insufficient sleep has been associated with poor glycemic control, increased insulin resistance, and heightened risks of hyperglycemia or hypoglycemia events [6–7]. Sleep disturbances, particularly those exacerbated by Ramadan fasting, could play a pivotal role in disrupting metabolic control, emphasizing the need for better understanding and management of sleep-related challenges in this population [7]. Previous studies have shown that Ramadan fasting can result in both hyperglycemia and hypoglycemia, posing risks for patients with diabetes [8]. Moreover, the alterations in circadian rhythms and sleep-wake cycles during Ramadan can further complicate the management of type 2 diabetes, leading to increased fatigue and sleep disturbances [9]. Therefore, it is imperative to understand the impact of Ramadan fasting on glycemic control and associated symptoms in this population.

Fatigue is a common and debilitating symptom experienced by individuals with type 2 diabetes, often exacerbated by poor glycemic control [10–11]. It can significantly impair daily

functioning and quality of life, making it a critical aspect of diabetes management. The relationship between glycemic fluctuations and fatigue is complex, involving various physiological and psychological mechanisms. Hyperglycemia can lead to osmotic diuresis and dehydration, resulting in fatigue, while hypoglycemia can cause neuroglycopenic symptoms such as weakness and lethargy [12]. To evaluate fatigue effectively, this study employed the Indonesian version of the Multidimensional Fatigue Inventory-20 (IMFI-20), a validated tool designed to assess fatigue across various dimensions. This inclusion provides a robust theoretical framework for understanding fatigue among individuals with diabetes [13]. Additionally, sleep disturbances, which are prevalent among individuals with type 2 diabetes, can further aggravate fatigue and disrupt overall metabolic control [14].

Dietary adjustments during Ramadan play an essential role in managing diabetes. The incorporation of low glycemic index (GI) foods, lean proteins, and fiber-rich options has been shown to improve glycemic control and sustain energy levels during fasting periods [15–16]. Such dietary strategies may also help mitigate fatigue and reduce the risk of extreme glycemic fluctuations. This study aimed to investigate the prevalence of glycemic fluctuations, fatigue, and sleep disturbances during Ramadan and to examine the factors associated with hypoglycemia and hyperglycemia events during this period.

Understanding the interplay between glycemic control, fatigue, and sleep disturbances during Ramadan is crucial for developing tailored management strategies for individuals with type 2 diabetes who choose to fast. While several studies have investigated the effects of Ramadan fasting on glycemic control, there is a paucity of research focusing on the concurrent impact on fatigue and sleep disturbances. By conducting a cross-sectional study, this research seeks to fill this gap in the literature and provide insights into the challenges faced by patients with diabetes during Ramadan. The findings of this study could inform healthcare providers in offering comprehensive guidance and support to patients with type 2 diabetes, ensuring safer fasting practices and improved overall well-being during Ramadan [17].

## 2  Methods

### 2.1  Study design and setting

This study was a cross-sectional investigation utilizing convenience sampling, conducted at a diabetes management center in Indonesia during the Ramadan fasting period from (08/03/2024) until (20/04/2024). The research received approval from the Joint Institutional Review Board of the Ethical Committee of Medical Research at the Faculty of Dentistry, University Jember (No.2458/UN25.8/KEPK/DL/2024). Written informed consent was obtained from all participants who agreed to take part in the survey. Additionally, the study adhered to the Strengthening the Reporting of Observational Studies in Epidemiology (STROBE) guidelines [18] (S1 Table).

### 2.2  Study populations

Participants were recruited from a diabetes management center in Indonesia. We utilized elements for risk calculation and the suggested risk score for individuals with diabetes mellitus who wish to fast during Ramadan [19]. Only those with a low to moderate risk score, and who chose to fast during Ramadan, were included in the study. Additionally, participants were required to be between 17 (the legal age for providing informed consent in Indonesia) and 65 years old, and to own a mobile phone. Individuals who could not read or write Indonesian, or who had been diagnosed with cognitive impairment, a psychiatric disorder, or cancer prior to the study, were excluded.

### 2.3 Outcomes

**2.3.1 Blood glucose levels.** Glucometers were used by the participants to monitor their own blood glucose levels twice daily (during the day and two hours after break fasting) during Ramadan [19]. The participants were invited to join a WhatsApp group to report their blood glucose levels each day during Ramadan. Blood glucose levels below 70 mg/dl were classified as hypoglycemia, and those above 200 mg/dl were considered hyperglycemia. Participants who experienced hypoglycemia were asked to break their fast during the period of fasting. Furthermore, HbA1c levels before Ramadan were collected from the medical records of the participants.

**2.3.2 Fatigue.** The fatigue levels of participants during Ramadan were measured using the Indonesian version of the Multidimensional Fatigue Inventory-20 (IMFI-20) [13]. The IMFI-20 consists of 20 items across 4 subscales: general and physical fatigue, reduced motivation, reduced activity, and mental fatigue. Each subscale includes 4 items rated on a 5-point Likert scale, ranging from 1 (strongly agree) to 5 (strongly disagree). We reverse-scored 10 positively worded items (items 2, 5, 9, 10, 13, 14, 16, 17, 18, and 19). The score for each subscale (ranging from 4 to 20 points) was the sum of its item scores, and the total fatigue score (ranging from 20 to 100 points) was the sum of the subscale scores, with higher scores indicating greater fatigue. The IMFI-20 was utilized due to its demonstrated reliability and validity in assessing specific aspects of fatigue in Indonesian-speaking patients with type 2 diabetes (Cronbach's α = 0.92).

**2.3.3 Sleep Quality.** Sleep quality among individuals with type 2 diabetes during Ramadan fasting was assessed using the Pittsburgh Sleep Quality Index (PSQI). The PSQI measures self-reported sleep quality and disturbances over the past month. It consists of 19 items across 7 dimensions: (1) subjective sleep quality, (2) sleep latency, (3) sleep duration, (4) sleep efficiency, (5) sleep disturbances, (6) use of sleeping medication, and (7) daytime dysfunction. Items are rated on a 4-point Likert scale ranging from 0 to 3, with an overall score range of 0 to 20. A score below 5 indicates good sleep quality [20]. The Indonesian version of the PSQI has proven to be highly valid and reliable for assessing sleep quality in Indonesian-speaking populations (Cronbach's α = 0.72) [21]. Furthermore, we have determined that the optimal sleep duration during Ramadan is between 7 and 8 hours [22].

**2.3.4 Demographic and disease characteristics and self-reported fatigue and sleep quality during Ramadan fasting.** Predesigned information sheets were employed to gather data on demographic and disease characteristics, including age, sex, education level, marital status, income level, and current diabetes treatment. Additionally, we included questions regarding the participants' experiences of fatigue and sleep quality, as well as the duration of these experiences during Ramadan fasting in a healthcare setting (S2 Table). Additionally, we collected data on participants' dietary habits during Ramadan fasting, encompassing aspects such as meal timing, frequency, and overall dietary patterns (with detailed definitions provided in S3 and S4 Table).

### 2.4 Data collection

Prior to Ramadan fasting, we invited potential participants at the diabetes management center to undergo a fasting risk assessment [18] and inquired about their intention to fast. Informed consent was obtained from those who agreed to participate. We also collected the WhatsApp numbers of all participants and created a WhatsApp group for reporting blood glucose levels during Ramadan. In the week following Ramadan, we invited participants back to the diabetes management center to complete the questionnaires and demographic form again. All participants who completed the study received a small gift.

### 2.5 Statistical analysis

All analyses were conducted using SPSS version 24.0 (IBM, Armonk, NY, USA), with a P value of < 0.05 considered statistically significant. Demographic and disease characteristics were presented as means and standard deviations for continuous variables and as numbers and percentages for categorical variables. After consulting with the physicians, we assessed several factors associated with hypoglycemia and hyperglycemia events during Ramadan fasting, which included type of medications, comorbidity, HbA1c levels before Ramadan, fatigue, and sleep quality, sleep duration, and dietary habits during Ramadan. The Chi-square test was employed to evaluate the associations between these factors and hypoglycemia and hyperglycemia events. An independent t-test was used to examine the relationship between the total IMFI-20 score and its domains with hypoglycemia and hyperglycemia events during Ramadan.

## 3 Results

### 3.1 General characteristics of the participants in the health care setting.

A total of 88 patients with type 2 diabetes (mean age of 52.7 years) participated in this study. The majority were female (68.2%) and married (63.6%). Approximately 89.8% of the participants were solely on oral hypoglycemic agents (OHA), with 58% having comorbid conditions. About 58% of the participants fasted for all 30 days of Ramadan, while 22.7% fasted for 15–29 days. The mean of HbA1c level before Ramadan was 6.7 mg/dl. Detailed participant characteristics are presented in Table 1.

### 3.2 Factor associated with glycemic fluctuations during Ramadan

Table 2 presents that the prevalence of hypoglycemia and hyperglycemia in type 2 diabetes during Ramadan was 21.6% and 30.6%, respectively. Additionally, 30.7% of participants experienced fatigue during Ramadan fasting, and 40.9% reported poor sleep quality during this period. The HbA1c levels of participants before Ramadan were significantly associated with

**Table 1. Participant characteristics and diseases (N = 88).**

| Variables | Total | |
|---|---|---|
| Age | 52.7 | (5.5) |
| Female | 60 | (68.2) |
| Diabetes duration (years) | 5.7 | (2.5) |
| Junior high school and above | 52 | (59.1) |
| Married | 56 | (63.6) |
| Having comorbidity | 51 | (58.0) |
| Medications | | |
| OHA alone | 79 | (89.8) |
| OHA combined with Insulin | 9 | (10.2) |
| Duration of fasting during Ramadan | | |
| 1-14 days | 17 | (19.3) |
| 15-29 days | 20 | (22.7) |
| 30 days | 51 | (58.0) |
| HbA1c (%) before Ramadan | 6.7 | (0.5) |

Continuous variables are presented in mean and standard deviation

Categorical variables are presented in number (%)

**Table 2. Relationship between glycemic fluctuations with factors during Ramadan fasting.**

| Groups | Total | | Hypoglycemia (n = 19) | | Non Hypoglycemia (n = 69) | | P | Hyperglycemia (n = 27) | | Non Hyperglycemia (n = 61) | | P |
|---|---|---|---|---|---|---|---|---|---|---|---|---|
| | n | % | n | % | n | % | | n | % | n | % | |
| Medications | | | | | | | 0.96 | | | | | 0.56 |
| OHA alone | 79 | (89.8) | 17 | (89.5) | 62 | (89.9) | | 25 | (92.6) | 54 | (88.5) | |
| OHA combined with Insulin | 9 | (10.2) | 2 | (22.2) | 7 | (10.1) | | 2 | (7.4) | 7 | (11.5) | |
| Comorbidity | | | | | | | 0.60 | | | | | 0.76 |
| No | 37 | (42.0) | 7 | (36.8) | 30 | (43.5) | | 12 | (44.4) | 25 | (41.0) | |
| Yes | 51 | (58.0) | 12 | (63.2) | 39 | (56.5) | | 15 | (55.6) | 36 | (59.0) | |
| HbA1c before Ramadan | | | | | | | **0.03** | | | | | **<0.001** |
| <7.5% | 74 | (84.1) | 19 | (100) | 55 | (79.7) | | 13 | (48.1) | 61 | (100) | |
| 7.5%-9.0% | 14 | (15.9) | 0 | (0) | 14 | (20.3) | | 14 | (51.9) | 0 | (0) | |
| Fatigue | | | | | | | **<0.001** | | | | | **0.02** |
| No | 61 | (69.3) | 5 | (26.3) | 56 | (81.2) | | 14 | (51.9) | 47 | (77.0) | |
| Yes | 27 | (30.7) | 14 | (73.7) | 13 | (18.8) | | 13 | (48.1) | 14 | (23.0) | |
| PSQI | | | | | | | 0.35 | | | | | **<0.001** |
| Good sleep quality (≤5) | 52 | (59.1) | 13 | (68.4) | 39 | (56.5) | | 8 | (29.6) | 44 | (72.1) | |
| Poor sleep quality (>5) | 36 | (40.9) | 6 | (31.6) | 30 | (43.5) | | 19 | (70.4) | 17 | (27.9) | |
| Sleep duration | | | | | | | 0.43 | | | | | **0.01** |
| < 7 hours | 25 | (28.4) | 7 | (36.8) | 18 | (26.1) | | 10 | (37.1) | 15 | (24.6) | |
| 7 – 8 hours | 48 | (54.5) | 7 | (36.8) | 41 | (59.4) | | 8 | (29.6) | 40 | (65.6) | |
| > 8 hours | 15 | (17.1) | 5 | (26.4) | 10 | (14.5) | | 9 | (33.3) | 6 | (9.8) | |
| Meal timing and frequency | | | | | | | **0.02** | | | | | **0.01** |
| Two meals | 16 | (18.2) | 13 | (68.4) | 3 | (4.4) | | 0 | (0) | 16 | (26.2) | |
| Three meals | 47 | (53.4) | 0 | (0) | 47 | (68.1) | | 7 | (25.9) | 40 | (65.6) | |
| Irregular meal pattern | 25 | (28.4) | 6 | (31.6) | 19 | (27.5) | | 20 | (74.1) | 5 | (8.2) | |
| Dietary patterns | | | | | | | **<0.001** | | | | | **<0.001** |
| Low glycemic foods | 49 | (55.7) | 4 | (21.1) | 45 | (65.2) | | 3 | (11.1) | 46 | (75.4) | |
| Random dietary habits | 39 | (44.3) | 15 | (78.9) | 24 | (34.8) | | 24 | (88.9) | 15 | (24.6) | |

OHA: Oral hypoglycemic agents. PSQI: Pittsburgh Sleep Quality Index.

both hypoglycemia (p = 0.03) and hyperglycemia cases (p < 0.001). Participants with higher HbA1c before Ramadan scores were more likely to experience hyperglycemia during Ramadan. Fatigue symptoms were significantly linked to both hypoglycemia and hyperglycemia events during Ramadan (p < 0.05), with most participants who reported fatigue also having abnormal blood glucose levels.

Sleep quality was significantly associated with hyperglycemia events (p < 0.0001). However, comorbidity and the type of medications did not show a significant association with hypoglycemia and hyperglycemia events during Ramadan (p > 0.05, Table 2). Additionally, sleep duration was found to have a significant association with hyperglycemia events (p = 0.01, Table 2). Participants who slept less than 7 hours or more than 8 hours were more likely to experience hyperglycemia events during Ramadan.

The timing and frequency of meals among participants during Ramadan were significantly associated to both hypoglycemia (p = 0.02) and hyperglycemia (p = 0.01, Table 2). Participants who consumed two meals during Ramadan were more likely to experience hypoglycemia. In contrast, those with an irregular meal pattern were more prone to experiencing

hyperglycemia during Ramadan. Moreover, dietary patterns were strongly associated to both hypoglycemia and hyperglycemia (both p < 0.0001, Table 2). Participants who followed a low GI food pattern during Ramadan experienced fewer hypoglycemia and hyperglycemic episodes than those with random dietary habits.

### 3.3 Glycemic fluctuations, dietary habits, and IMFI-20 during Ramadan

Table 3 indicates that both hypoglycemia and hyperglycemia were significantly associated with the total score of the IMFI-20 (p < 0.05). Hypoglycemia was associated to all three domains of the IMFI-20, including general/physical fatigue, mental fatigue, and reduced activity (p < 0.05), whereas hyperglycemia was only significantly associated with general/physical fatigue during Ramadan, as detailed in Table 3.

In addition, the timing and frequency of meals and dietary patterns during Ramadan were significantly correlated with the overall score of the IMFI-20 and general/physical fatigue (all p < 0.05). Meal timing and frequency also showed significant associations the reduced activity (p < 0.05, Table 3).

### 3.4 Fatigue and sleep quality among type 2 diabetes during Ramadan in health care settings.

Concerning fatigue and sleep quality among individuals with type 2 diabetes during Ramadan (as shown in Tables 4 and 5), 46.7% of participants who experienced fatigue and 75% of those with poor sleep quality during Ramadan fasting rarely or never discussed these issues with their physicians. Additionally, participants with fatigue (81.5%) and poor sleep quality (86.1%) were rarely or never treated by their physicians.

## 3 Discussion

This recent study found that hypoglycemia and hyperglycemia are prevalent among individuals with type 2 diabetes during Ramadan fasting. Additionally, some of the participants experienced fatigue and poor sleep quality during this period. The HbA1c level before Ramadan and

**Table 3. Glycemic fluctuations, dietary habits, and IMFI-20 during Ramadan.**

| Groups | n | IMFI-20 | General/physical fatigue | Mental fatigue | Reduced Activity | Reduced Motivation |
|---|---|---|---|---|---|---|
| Hypoglycemia | | | | | | |
| No | 69 | 53.3 ± 12.2[*] | 24.4 ± 7.5[*] | 10.1 ± 1.8[*] | 8.9 ± 2.5[*] | 9.6 ± 4.3 |
| Yes | 19 | 60.0 ± 11.5[*] | 27.2 ± 7.8[*] | 11.1 ± 2.2[*] | 10.7 ± 2.4[*] | 11.2 ± 4.6 |
| Hyperglycemia | | | | | | |
| No | 61 | 52.8 ± 10.9[*] | 23.8 ± 6.6[*] | 10.2 ± 2.0 | 9.3 ± 2.7 | 9.5 ± 3.7 |
| Yes | 27 | 58.9 ± 14.2[*] | 28.3 ± 8.7[*] | 10.5 ± 1.6 | 9.4 ± 2.3 | 11.0 ± 5.4 |
| Meal timing and frequency | | | | | | |
| Two meals | 16 | 59.9 ± 11.2[*] | 26.5 ± 6.2[*] | 10.9 ± 2.1 | 10.3 ± 2.6[*] | 10.6 ± 4.4 |
| Three meals | 47 | 52.3 ± 11.1[*] | 24.1 ± 6.7[*] | 10.3 ± 1.9 | 8.2 ± 2.5[*] | 9.6 ± 3.8 |
| Irregular meal pattern | 25 | 59.5 ± 10.5[*] | 28.1 ± 8.1[*] | 10.2 ± 2.0 | 9.5 ± 2.3[*] | 10.9 ± 5.1 |
| Dietary patterns | | | | | | |
| Low glycemic foods | 49 | 51.6 ± 10.4[*] | 22.7 ± 6.5[*] | 10.1 ± 2.1 | 9.2 ± 2.5 | 9.6 ± 3.5 |
| Random dietary habits | 39 | 58.1 ± 13.9[*] | 28.1 ± 8.6[*] | 10.4 ± 1.5 | 9.4 ± 2.4 | 10.6 ± 5.5 |

Values are presented as mean and standard deviation.

[*] p < 0.05.

**Table 4. Healthcare service usage for addressing fatigue among people with type 2 diabetes during Ramadan.**

| Questions | Total | | Fatigued (n = 27) | | Non-Fatigued (n = 61) | |
|---|---|---|---|---|---|---|
| | n | % | n | % | n | % |
| How long have you felt fatigue during Ramadan? (day), mean (SD) | 1.8 | 2.9 | 5.9 | 2.0 | 00.0 | 0.0 |
| Have you ever discussed "the feeling of tired" or "fatigue" with your physician during Ramadan? | | | | | | |
| Never | 69 | (78.4) | 8 | (9.6) | 61 | (100) |
| Rarely | 10 | (11.4) | 10 | (37.1) | 0 | (0) |
| Sometimes | 5 | (5.7) | 5 | (18.5) | 0 | (0) |
| Often | 4 | (4.5) | 4 | (14.8) | 0 | (0) |
| Always | 0 | (0) | 0 | (0) | 0 | (0) |
| Whether your fatigue level has been measured by your physician during Ramadan? | | | | | | |
| Never | 75 | (85.2) | 14 | (51.9) | 61 | (100) |
| Rarely | 8 | (9.1) | 8 | (29.6) | 0 | (0) |
| Sometimes | 5 | (5.7) | 5 | (18.5) | 0 | (0) |
| Often | 0 | (0) | 0 | (0) | 0 | (0) |
| Always | 0 | (0) | 0 | (0) | 0 | (0) |
| Has your fatigue been treated by physician during Ramadan? | | | | | | |
| Never | 73 | (82.9) | 12 | (44.4) | 61 | (100) |
| Rarely | 10 | (11.4) | 10 | (37.1) | 0 | (0) |
| Sometimes | 5 | (5.7) | 5 | (18.5) | 0 | (0) |
| Often | 0 | (0) | 0 | (0) | 0 | (0) |
| Always | 0 | (0) | 0 | (0) | 0 | (0) |

**Table 5. Healthcare service usage for addressing sleep quality among people with type 2 diabetes during Ramadan.**

| Questions | Total | | Poor Sleep (n = 36) | | Good Sleep (n = 52) | |
|---|---|---|---|---|---|---|
| | n | % | n | % | n | % |
| How long have you had sleep disturbance during Ramadan? (day), mean (SD) | 2.2 | 2.8 | 5.3 | 1.6 | 0.0 | 0.0 |
| Have you ever discussed "the experiencing of sleep disturbance" with your physician during Ramadan? | | | | | | |
| Never | 60 | (68.2) | 10 | (27.8) | 50 | (96.2) |
| Rarely | 19 | (21.6) | 17 | (47.2) | 2 | (3.8) |
| Sometimes | 5 | (5.7) | 5 | (13.9) | 0 | (0) |
| Often | 4 | (4.5) | 4 | (11.1) | 0 | (0) |
| Always | 0 | (0) | 0 | (0) | 0 | (0) |
| Has your sleep quality level been measured by your physician during Ramadan? | | | | | | |
| Never | 62 | (70.4) | 17 | (47.2) | 45 | (86.5) |
| Rarely | 21 | (23.9) | 14 | (38.9) | 7 | (13.5) |
| Sometimes | 5 | (5.7) | 5 | (13.9) | 0 | (0) |
| Often | 0 | (0) | 0 | (0) | 0 | (0) |
| Always | 0 | (0) | 0 | (0) | 0 | (0) |
| Has your sleep disturbance been treated by a physician during Ramadan? | | | | | | |
| Never | 58 | (65.9) | 17 | (47.2) | 45 | (86.5) |
| Rarely | 25 | (28.4) | 14 | (38.9) | 7 | (13.5) |
| Sometimes | 5 | (5.7) | 5 | (13.9) | 0 | (0) |
| Often | 0 | (0) | 0 | (0) | 0 | (0) |
| Always | 0 | (0) | 0 | (0) | 0 | (0) |

fatigue were significantly associated with the occurrence of both hypoglycemia and hyperglycemia during Ramadan. Poor sleep quality was also a significant factor associated with hyperglycemia during this time.

The findings of our study indicate that hyperglycemia was more prevalent than hypoglycemia during Ramadan. These results highlight the dual challenge faced by individuals with glucose regulation issues during Ramadan fasting [23]. Previous studies have documented similar challenges in patients with type 1 diabetes [24], but our study adds to the growing body of evidence by quantifying the extent of these issues in type 2 diabetes. The high prevalence underscores the need for tailored medical advice and monitoring for individuals with prediabetes or diabetes during Ramadan.

Our analysis revealed that fatigue and poor sleep quality are prevalent among the fasting individuals. These factors are critical as they not only impact daily functioning but also appear to be linked to glucose regulation. The association between fatigue and both hypoglycemia and hyperglycemia suggests that physical and mental exhaustion could influence metabolic control during Ramadan. This is consistent with the understanding that stress and lack of rest can exacerbate glucose fluctuations [25], emphasizing the importance of managing overall health and well-being during fasting.

Additionally, our findings revealed that hypoglycemia and hyperglycemia was significantly associated with the total score of IMFI-20, highlighting the broad impact of fatigue on individuals experiencing glycemic fluctuations. Hypoglycemia was also linked to specific domains of the IMFI-20, including general/physical fatigue, mental fatigue, and reduced activity. Similarly, hyperglycemia was found to be associated with general/physical fatigue. These associations underscore the critical interplay between glucose regulation and various dimensions of fatigue, emphasizing the need for comprehensive management strategies during Ramadan fasting.

Moreover, we recommend aiming for 7–8 hours of sleep per night during Ramadan as an optimal sleep duration. This finding aligns with prior research, which has highlighted that obtaining 7–8 hours of sleep per night is associated with better regulation of blood glucose levels [22]. Encouraging individuals with diabetes to achieve this sleep duration during Ramadan could help stabilize blood glucose levels and reduce the risk of both hypoglycemia and hyperglycemia. Inadequate sleep can disrupt hormonal balance and insulin action, increasing the risk of blood glucose fluctuations [26]. Interventions focused on improving sleep hygiene such as setting a regular sleep schedule, avoiding caffeine, and reducing screen time before bed could be beneficial in managing glucose levels during Ramadan.

Importantly, our study identified HbA1c levels prior to Ramadan as significant factors associated with hypoglycemia and hyperglycemia events during fasting. This finding suggests that individuals with higher baseline HbA1c levels are more susceptible to glucose fluctuations, which could be attributed to their overall poorer glycemic control. These insights are crucial for healthcare providers in developing personalized strategies to minimize the risks of hypoglycemia and hyperglycemia for fasting individuals.

Sleep quality also emerged as a significant factor associated with hyperglycemia events during Ramadan. Poor sleep quality can lead to hormonal imbalances that affect insulin sensitivity and glucose metabolism [27], thereby increasing the risk of hyperglycemia. This relationship underscores the importance of adequate and quality sleep in managing glucose levels. Interventions aimed at improving sleep hygiene could, therefore, be beneficial for individuals fasting during Ramadan. Future research should explore targeted interventions to enhance sleep quality and reduce fatigue as part of comprehensive diabetes management during fasting periods.

In addition to these findings, our study revealed that meal timing and frequency during Ramadan are significantly associated with glycemic fluctuations. Participants who followed a three-meal pattern (i.e., Suhoor, Iftar, and a post-Iftar snack) experienced better glycemic control compared to those who consumed only two meals (i.e., Iftar and Suhoor) or had an irregular meal pattern. The three-meal pattern appeared to help stabilize blood glucose levels by spreading food intake more evenly throughout the day, reducing the likelihood of both hypoglycemia and hyperglycemia events. Irregular meal patterns, in contrast, may lead to more dramatic fluctuations in blood glucose, likely due to inconsistent insulin secretion and delayed absorption of nutrients. These findings underscore the importance of structured meal timing for individuals with diabetes during Ramadan. A consistent eating schedule may help mitigate the risk of glycemic fluctuations and improve overall metabolic control during fasting.

Furthermore, our study found that dietary patterns also play a critical role in regulating glycemic fluctuations during Ramadan. Participants who adhered to a low GI pattern focusing on foods that are known to have minimal impact on blood glucose levels experienced fewer instances of both hypoglycemia and hyperglycemia compared to those with more random eating habits. This suggests that following a low GI dietary pattern during Ramadan may contribute to more stable blood glucose levels, reducing the risk of glycemic fluctuations. These findings underscore the importance of educating individuals with diabetes on the benefits of low GI foods, especially during Ramadan, to help manage glucose levels more effectively. The impact of dietary habits on glucose regulation highlights the need for tailored dietary advice as part of a comprehensive management plan for individuals fasting during Ramadan.

## 4.1 Limitation and strengths

This study has several limitations. Firstly, the participant pool was limited to individuals residing in urban areas, potentially restricting the broader applicability of the findings. Secondly, the cross-sectional design of this study makes it challenging to establish causality between exposure and outcomes. Despite these limitations, this research is pioneering in investigating the relationship between fatigue, sleep quality, and glycemic fluctuations among individuals with type 2 diabetes during Ramadan fasting.

## 4 Conclusion

This study shows that hypoglycemia and hyperglycemia are common among people with type 2 diabetes during Ramadan fasting. Many participants also experienced fatigue and poor sleep quality. We found that HbA1c levels before Ramadan and fatigue were key factors linked to both hypoglycemia and hyperglycemia. Poor sleep quality was also linked to higher rates of hyperglycemia. These results highlight the importance of targeted care to reduce glycemic fluctuations and improve overall health during Ramadan fasting. To manage these issues, we suggest healthcare providers encourage patients with type 2 diabetes to get 7–8 hours of sleep per night during Ramadan. This sleep duration can help stabilize blood glucose levels. Furthermore, consuming three meals a day comprising Suhoor, Iftar, and a post-Iftar snack while following a low glycemic pattern is beneficial for maintaining blood glucose stability and reducing the risk of both hypoglycemia and hyperglycemia.

We also recommend using tools like the IMFI-20 to track fatigue and the PSQI to assess sleep quality. These tools can help healthcare providers monitor symptoms and tailor interventions. Future research should explore the effects of dietary changes, sleep, and fatigue management on blood glucose control during Ramadan. Understanding how sleep, food, and glycemic control interact during fasting will help improve care for people with type 2 diabetes during Ramadan.

## Supporting Information

**S1 Table. STROBE Statement of cross-sectional studies.**
(DOCX)

**S2 Table. Additional questions related experience fatigue and sleep quality among type 2 diabetes during Ramadan fasting in health care settings.**
(DOCX)

**S3 Table. Meal timing and frequency during Ramadan fasting.**
(DOCX)

**S4 Table. Dietary patterns during Ramadan fasting.**
(DOCX)

**S1 Data. Data.**
(CSV)

## Acknowledgments

None.

## Author contributions

**Conceptualization:** Debby Syahru Romadlon.

**Data curation:** Debby Syahru Romadlon, Satwika Arya Pratama, Rudy Kurniawan.

**Formal analysis:** Debby Syahru Romadlon, Satwika Arya Pratama.

**Funding acquisition:** Debby Syahru Romadlon.

**Investigation:** Debby Syahru Romadlon, Satwika Arya Pratama, Rudy Kurniawan, Hsiao-Yean Chiu.

**Methodology:** Debby Syahru Romadlon, Rudy Kurniawan.

**Project administration:** Debby Syahru Romadlon, Satwika Arya Pratama, Rudy Kurniawan.

**Resources:** Debby Syahru Romadlon, Satwika Arya Pratama, Rudy Kurniawan, Safiruddin Al-Baqi.

**Software:** Debby Syahru Romadlon.

**Supervision:** Debby Syahru Romadlon, Hsiao-Yean Chiu, Hsuan-Ju Kuo.

**Validation:** Debby Syahru Romadlon, Hsiao-Yean Chiu, Emmanuel Ekpor, Faizul Hasan, Po-Jen Kung.

**Visualization:** Debby Syahru Romadlon.

**Writing – original draft:** Debby Syahru Romadlon, Satwika Arya Pratama, Rudy Kurniawan.

**Writing – review & editing:** Debby Syahru Romadlon.

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
