## [Decision Letter · Decision Letter 0]

25 Nov 2024

PONE-D-24-43792Glycemic Fluctuations, Fatigue, and Sleep Disturbances in Type 2 Diabetes During Ramadan Fasting: A Cross-Sectional StudyPLOS ONE

Dear Dr. Romadlon,

Thank you for submitting your manuscript to PLOS ONE. After careful consideration, we feel that it has merit but does not fully meet PLOS ONE’s publication criteria as it currently stands. Therefore, we invite you to submit a revised version of the manuscript that addresses the points raised during the review process.

We look forward to receiving your revised manuscript.

Kind regards,

Julio Alejandro Henriques Castro da Costa

Academic Editor

PLOS ONE

Journal Requirements:

2. Thank you for stating the following in your Competing Interests section: [The authors declare no conflicts of interest.]. Please complete your Competing Interests on the online submission form to state any Competing Interests. If you have no competing interests, please state "The authors have declared that no competing interests exist.", as detailed online in our guide for authors at http://journals.plos.org/plosone/s/submit-now This information should be included in your cover letter; we will change the online submission form on your behalf.

3. In this instance it seems there may be acceptable restrictions in place that prevent the public sharing of your minimal data. However, in line with our goal of ensuring long-term data availability to all interested researchers, PLOS’ Data Policy states that authors cannot be the sole named individuals responsible for ensuring data access (http://journals.plos.org/plosone/s/data-availability#loc-acceptable-data-sharing-methods). Data requests to a non-author institutional point of contact, such as a data access or ethics committee, helps guarantee long term stability and availability of data. Providing interested researchers with a durable point of contact ensures data will be accessible even if an author changes email addresses, institutions, or becomes unavailable to answer requests. Before we proceed with your manuscript, please also provide non-author contact information (phone/email/hyperlink) for a data access committee, ethics committee, or other institutional body to which data requests may be sent. If no institutional body is available to respond to requests for your minimal data, please consider if there any institutional representatives who did not collaborate in the study, and are not listed as authors on the manuscript, who would be able to hold the data and respond to external requests for data access? If so, please provide their contact information (i.e., email address). Please also provide details on how you will ensure persistent or long-term data storage and availability.

4. In the online submission form, you indicated that [Specific data files used in the analysis are available from the corresponding author upon reasonable request.]. All PLOS journals now require all data underlying the findings described in their manuscript to be freely available to other researchers, either 1. In a public repository, 2. Within the manuscript itself, or 3. Uploaded as supplementary information. This policy applies to all data except where public deposition would breach compliance with the protocol approved by your research ethics board. If your data cannot be made publicly available for ethical or legal reasons (e.g., public availability would compromise patient privacy), please explain your reasons on resubmission and your exemption request will be escalated for approval.

5. Please include captions for your Supporting Information files at the end of your manuscript, and update any in-text citations to match accordingly. Please see our Supporting Information guidelines for more information: http://journals.plos.org/plosone/s/supporting-information .

Reviewers' comments:

Reviewer's Responses to Questions

**Comments to the Author**

1. Is the manuscript technically sound, and do the data support the conclusions?

Reviewer #1: Partly

2. Has the statistical analysis been performed appropriately and rigorously? 

Reviewer #1: Yes

3. Have the authors made all data underlying the findings in their manuscript fully available?

Reviewer #1: No

4. Is the manuscript presented in an intelligible fashion and written in standard English?

Reviewer #1: Yes

5. Review Comments to the Author

Reviewer #1: Dear Author(s),

As a reviewer, I have carefully read your manuscript, titled "Glycemic Fluctuations, Fatigue, and Sleep Disturbances in Type 2 Diabetes During Ramadan Fasting: A Cross-Sectional Study." I appreciate your effort in conducting this important research, which addresses critical aspects of managing type 2 diabetes during Ramadan fasting. Below, I provide several suggestions that I believe would enhance the clarity, scientific rigor, and practical impact of your study.

---

Suggestions for Improvement:

1. Abstract:

Consider including specific recommendations for sleep duration to improve clarity for readers. For example, highlighting 7–8 hours as an optimal “cut-off” could offer a practical guideline.

Briefly mention dietary factors that may benefit glycemic control, such as low glycemic index foods, lean proteins, and healthy fats.

Clarify the use of the Multidimensional Fatigue Inventory-20 (IMFI-20) for assessing fatigue, which would strengthen the scientific credibility of your work.

2. Introduction:

Discuss the importance of adequate sleep for individuals with diabetes and its impact on blood glucose regulation. Including more background on sleep needs could enhance the relevance of your study.

Introduce the potential benefits of a low glycemic index diet, lean proteins, and fiber-rich foods for blood glucose control, especially during fasting periods.

Mention the use of IMFI-20 as a validated tool for assessing fatigue to provide a theoretical basis for including this measurement.

3. Methods:

Expand on your choice of fatigue and sleep assessment tools (IMFI-20 and PSQI) and their validation in the target population.

Clearly define an optimal sleep duration (such as 7–8 hours) as a benchmark to evaluate sleep adequacy in your participants.

Consider adding details about participants’ dietary habits and whether specific dietary recommendations were made, as this could impact blood glucose and fatigue outcomes.

4. Results:

Provide more detailed data on the relationship between sleep quality and blood glucose control, and clarify if specific sleep durations (like 7–8 hours) are associated with better glucose management.

Include information on how dietary habits (e.g., low glycemic index foods) affected blood glucose levels and fatigue to offer a more comprehensive understanding of these variables.

Analyze each dimension of the IMFI-20 separately to show the specific areas of fatigue most affected by blood glucose fluctuations.

5. Discussion:

Offer practical recommendations for optimal sleep, such as encouraging 7–8 hours per night to improve blood glucose stability during Ramadan.

Include dietary suggestions, particularly emphasizing the role of low glycemic index foods, lean proteins, and healthy fats in blood glucose management.

Provide a detailed analysis of fatigue dimensions using IMFI-20 results to better illustrate the physiological and psychological impact of Ramadan fasting on diabetic patients.

6. Conclusion:

Suggest specific recommendations on diet and sleep duration for diabetic patients during Ramadan to make the conclusion actionable.

Emphasize the importance of using tools like IMFI-20 and PSQI in routine diabetes management during fasting periods.

Consider proposing future research that focuses on the effects of various dietary interventions and sleep duration on blood glucose and fatigue during Ramadan.

---

Incorporating these suggestions would not only add depth to your findings but also make the results more accessible and applicable to clinical practice. I commend you for this important work and hope that these recommendations are helpful in enhancing your manuscript.

Thank you for the opportunity to review your study.

Best regards,

[Your Name]

[Your Affiliation]

6. PLOS authors have the option to publish the peer review history of their article (what does this mean? ). If published, this will include your full peer review and any attached files.

**Do you want your identity to be public for this peer review?** For information about this choice, including consent withdrawal, please see our Privacy Policy .

Reviewer #1: No

---

## [Author Response · Author response to Decision Letter 1]

10 Dec 2024

Reviewer #1: Dear Author(s),

As a reviewer, I have carefully read your manuscript, titled "Glycemic Fluctuations, Fatigue, and Sleep Disturbances in Type 2 Diabetes During Ramadan Fasting: A Cross-Sectional Study." I appreciate your effort in conducting this important research, which addresses critical aspects of managing type 2 diabetes during Ramadan fasting. Below, I provide several suggestions that I believe would enhance the clarity, scientific rigor, and practical impact of your study.

Response: Thank you for your kind words about our project. We have thoughtfully reviewed the feedback and made revisions to the paper based on your suggestions. Below, you will find our detailed responses addressing each of the comments. We hope these updates meet your expectations and adequately respond to the feedback. Thanks again for taking the time to review our work.

Suggestions for Improvement:

1. Abstract:

Consider including specific recommendations for sleep duration to improve clarity for readers. For example, highlighting 7–8 hours as an optimal “cut-off” could offer a practical guideline.

Briefly mention dietary factors that may benefit glycemic control, such as low glycemic index foods, lean proteins, and healthy fats.

Response: Thank you for the valuable feedbacks. We have provided the additional informations related to sleep duration and dietary habits including meal timing and frequency and dietary patterns among participants during Ramadan in the abstract. In the methods section as follow: “In addition, data on sleep duration, as well as dietary habits during Ramadan, were also collected”. In the result sections as follow: Furthermore, sleep duration was significantly related to hyperglycemia events (p = 0.01). Meal timing, frequency, and dietary patterns during Ramadan were also found to be significantly associated with both hypoglycemia and hyperglycemia (both p < 0.05)”. And in the conlusion part as follow: “We recommend that healthcare providers advise patients with type 2 diabetes to aim for 7–8 hours of sleep per night to help control blood glucose levels. Additionally, having three meals a day (Suhoor, Iftar, and a post-Iftar snack) with low glycemic index foods can help maintain stable blood glucose and prevent both hypoglycemia and hyperglycemia during Ramadan”. All changes are marked in red.

Clarify the use of the Multidimensional Fatigue Inventory-20 (IMFI-20) for assessing fatigue, which would strengthen the scientific credibility of your work.

Response: Thank you for the comments and suggestions. We have included information on the use of the IMFI-20 for assessing fatigue in the abstract section as follows:“Fatigue was thoroughly assessed using the Indonesian Multidimensional Fatigue Inventory-20 (IMFI-20)”. All changes are marked in red.

2. Introduction:

Discuss the importance of adequate sleep for individuals with diabetes and its impact on blood glucose regulation. Including more background on sleep needs could enhance the relevance of your study.

Response: Thank you for the comments and suggestions. We have added the informations regarding the importance of adequate sleep for individuals with diabetes in the introduction section as follow: “Adequate sleep is crucial for maintaining blood glucose regulation in individuals with diabetes [6]. Insufficient sleep has been associated with poor glycemic control, increased insulin resistance, and heightened risks of hyperglycemia or hypoglycemia events [6-7]. Sleep disturbances, particularly those exacerbated by Ramadan fasting, could play a pivotal role in disrupting metabolic control, emphasizing the need for better understanding and management of sleep-related challenges in this population [7]”. All changes are marked in red.

Introduce the potential benefits of a low glycemic index diet, lean proteins, and fiber-rich foods for blood glucose control, especially during fasting periods.

Response: Thank you for the comments and suggestions. We have added the informations about that in the introduction as follow: “The incorporation of low glycemic index (GI) foods, lean proteins, and fiber-rich options has been shown to improve glycemic control and sustain energy levels during fasting periods [15-16]. Such dietary strategies may also help mitigate fatigue and reduce the risk of extreme glycemic fluctuations”. All changes are marked in red.

Mention the use of IMFI-20 as a validated tool for assessing fatigue to provide a theoretical basis for including this measurement.

Response: Thank you for the comments and suggestions. We have added the informations the use of IMFI-20 in this study in the introduction section as follow:“To evaluate fatigue effectively, this study employed the Indonesian version of the Multidimensional Fatigue Inventory-20 (IMFI-20) , a validated tool designed to assess fatigue across various dimensions. This inclusion provides a robust theoretical framework for understanding fatigue among individuals with diabetes [13]”. All changes are marked in red.

3. Methods:

Expand on your choice of fatigue and sleep assessment tools (IMFI-20 and PSQI) and their validation in the target population.

Response: Thank you for your valuable feedback. To begin with, we selected the Indonesian version of the Multidimensional Fatigue Inventory-20 (IMFI-20) because it provides a thorough assessment of fatigue across four domains: general and physical fatigue, mental fatigue, reduced activity, and reduced motivation. Furthermore, the IMFI-20 has demonstrated reliability and validity in evaluating specific aspects of fatigue in Indonesian-speaking patients with type 2 diabetes. We have included additional details in the methods section as follows:“The IMFI-20 was utilized due to its demonstrated reliability and validity in assessing specific aspects of fatigue in Indonesian-speaking patients with type 2 diabetes (Cronbach’s α = 0.92)”.

In addition, we have selected PSQI in our study is widely recognized as a valid and reliable tool to assess sleep quality. PSQI also well know instrument to assess sleep quality in patients with chronic diseases including type 2 diabetes. Furthermore, the Indonesian version of PSQI showed the high validity and reliability to assess the sleep quality in Indonesian speaking populations. We have included additional details in the methods section as follows: “The Indonesian version of the PSQI has proven to be highly valid and reliable for assessing sleep quality in Indonesian-speaking populations”. All changes are marked in red.

Clearly define an optimal sleep duration (such as 7–8 hours) as a benchmark to evaluate sleep adequacy in your participants.

Response: Thank you for the valuable feedback and suggestions. We have added information more details about sleep duration in methodology section section as follows: “Furthermore, we have determined that the optimal sleep duration during Ramadan is between 7 and 8 hours [22]”. All changes are marked in red.

Consider adding details about participants’ dietary habits and whether specific dietary recommendations were made, as this could impact blood glucose and fatigue outcomes.

Response: Thank you for your valuable comments and suggestions. Concerning the dietary habits of the participants, we have included details on the meal timing and frequency and dietary patterns during Ramadan. This information has been added to the methodology section and supplementary material table 3 and table 4 as follows: “Additionally, we collected data on participants’ dietary habits during Ramadan fasting, encompassing aspects such as meal timing, frequency, and overall dietary patterns (with detailed definitions provided in S3 and S4 Table)”. All changes are marked in red.

4. Results:

Provide more detailed data on the relationship between sleep quality and blood glucose control, and clarify if specific sleep durations (like 7–8 hours) are associated with better glucose management.

Response: Thank you for the comments and suggestions. With regard to the relationship between sleep duration and blood glucose levels, we have included additional information in the results section as follows: “Additionally, sleep duration was found to have a significant association with hyperglycemia events (p = 0.01, Table 2). Participants who slept less than 7 hours or more than 8 hours were more likely to experience hyperglycemia events during Ramadan”. We have also included the sleep duration variable in Table 2. All changes are marked in red.

Include information on how dietary habits (e.g., low glycemic index foods) affected blood glucose levels and fatigue to offer a more comprehensive understanding of these variables.

Response: Thank you for the valuable feedback. Regarding to the dietary habis in this study, we have added the informations about meal timing and frequency and dietary patterns among participants during Ramadan. We have provided the relationship between dietary habits and incidence of hypoglycemia and hyperglycemia in the result section as follow:“The timing and frequency of meals among participants during Ramadan were significantly associated to both hypoglycemia (p = 0.02) and hyperglycemia (p = 0.01, Table 2). Participants who consumed two meals during Ramadan were more likely to experience hypoglycemia. In contrast, those with an irregular meal pattern were more prone to experiencing hyperglycemia during Ramadan. Moreover, dietary patterns were strongly associated to both hypoglycemia and hyperglycemia (both p < 0.0001, Table 2). Participants who followed a low GI food pattern during Ramadan experienced fewer hypoglycemia and hyperglycemic episodes than those with random dietary habits”. All changes are marked in red.

Furthermore, we have added the association between dietary habits and the IMFI-20 in the result section as follow: “In addition, the timing and frequency of meals and dietary patterns during Ramadan were significantly correlated with the overall score of the IMFI-20 and general/physical fatigue (all p < 0.05). Meal timing and frequency also showed significant associations the reduced activity (p < 0.05, Table 3)”. All changes are marked in red.

Analyze each dimension of the IMFI-20 separately to show the specific areas of fatigue most affected by blood glucose fluctuations.

Response: Thank you for the valuable feedbacks. We have presented the relationship between glycemic fluctuations and the total IMFI-20 score, as well as its domains, in the results section as follows:“Table 3 indicates that both hypoglycemia and hyperglycemia were significantly associated with the total score of the IMFI-20 (p < 0.05). Hypoglycemia was associated to all three domains of the IMFI-20, including general/physical fatigue, mental fatigue, and reduced activity (p < 0.05), whereas hyperglycemia was only significantly associated with general/physical fatigue during Ramadan, as detailed in Table 3”. All changes are marked in red.

5. Discussion:

Offer practical recommendations for optimal sleep, such as encouraging 7–8 hours per night to improve blood glucose stability during Ramadan.

Response: Thank you for the comments and suggestions. We have added the informations regarding to the recommendations for optimal sleep duration during Ramadan fasting in the discussion section as follow: “Moreover, we recommend aiming for 7–8 hours of sleep per night during Ramadan as an optimal sleep duration. This finding aligns with prior research, which has highlighted that obtaining 7-8 hours of sleep per night is associated with better regulation of blood glucose levels [22]. Encouraging individuals with diabetes to achieve this sleep duration during Ramadan could help stabilize blood glucose levels and reduce the risk of both hypoglycemia and hyperglycemia. Inadequate sleep can disrupt hormonal balance and insulin action, increasing the risk of blood glucose fluctuations [26]. Interventions focused on improving sleep hygiene such as setting a regular sleep schedule, avoiding caffeine, and reducing screen time before bed could be beneficial in managing glucose levels during Ramadan”. All changes are marked in red.

Include dietary suggestions, particularly emphasizing the role of low glycemic index foods, lean proteins, and healthy fats in blood glucose management.

Response: Thank you for the comments and suggestions. We have added the informations about dietary habits among participants during Ramadan in the discussion part as follow: “In addition to these findings, our study revealed that meal timing and frequency during Ramadan are significantly associated with glycemic fluctuations. Participants who followed a three-meal pattern (i.e., Suhoor, Iftar, and a post-Iftar snack) experienced better glycemic control compared to those who consumed only two meals (i.e., Iftar and Suhoor) or had an irregular meal pattern. The three-meal pattern appeared to help stabilize blood glucose levels by spreading food intake more evenly throughout the day, reducing the likelihood of both hypoglycemia and hyperglycemia events. Irregular meal patterns, in contrast, may lead to more dramatic fluctuations in blood glucose, likely due to inconsistent insulin secretion and delayed absorption of nutrients. These findings underscore the importance of structured meal timing for individuals with diabetes during Ramadan. A consistent eating schedule may help mitigate the risk of glycemic fluctuations and improve overall metabolic control during fasting.

Furthermore, our study found that dietary patterns also play a critical role in regulating glycemic fluctuations during Ramadan. Participants who adhered to a low GI pattern focusing on foods that are known to have minimal impact on blood glucose levels experienced fewer instances of both hypoglycemia and hyperglycemia compared to those with more random eating habits. This suggests that following a low GI dietary pattern during Ramadan may contribute to more stable blood glucose levels, reducing the risk of glycemic fluctuations. These findings underscore the importance of educating individuals with diabetes on the benefits of low GI foods, especially during Ramadan, to help manage glucose levels more effectively. The impact of dietary habits on glucose regulation highlights the need for tailored dietary advice as part of a comprehensive management plan for individuals fasting during Ramadan”. All changes are marked in red.

Provide a detailed analysis of fatigue dimensions using IMFI-20 results to better illustrate the physiological and psychological impact of Ramadan fasting on diabetic patients.

Response: Thank you for the commenst and suggestions. We have included the information related to the IMFI-20 domains in the discussion section as follows:“Additionally, our findings revealed that hypoglycemia and hyperglycemia was significantly associated with the total score of IMFI-20, highlighting the broad impact of fatigue on individuals experiencing glycemic fluctuations. Hypoglycemia was also linked to specific domains of the IMFI-20, including general/physical fatigue, mental fatigue, and reduced activity. Similarly, hyperglycemia was found to be associated with general/physical fatigue. These associations underscore the critical interplay between glucose regulation and various dimensions of fatigue, emphasizing the need for comprehensive management strategies during Ramadan fasting”. All changes are marked in red.

6. Conclusion:

Suggest specific recommendations on diet and sleep duration for diabetic patients during Ramadan to make the conclusion actionable.

Response: Thank you for the comment and suggestion. We have provided the recommendations on dietary habits and sleep duration for patients with diabetes during Ramadan in the conclusion part as follow:“To manage these issues, we suggest healthcare providers encourage patients with type 2 diabetes to get 7–8 hours of sleep per night during Ramadan. This sleep duration can help stabilize blood glucose levels. Furthermore, consuming three meals a day comprising Suhoor, Iftar, and a post-Iftar snack while following a low glycemic pattern is beneficial for maintaining blood glucose stability and reducing the risk of both hypoglycemia and hyperglycemia”. All changes are marked in red.

Emphasize the importance of using to

---

## [Decision Letter · Decision Letter 1]

17 Dec 2024

Glycemic Fluctuations, Fatigue, and Sleep Disturbances in Type 2 Diabetes During Ramadan Fasting: A Cross-Sectional Study

PONE-D-24-43792R1

Dear Dr. Romadlon,

We’re pleased to inform you that your manuscript has been judged scientifically suitable for publication and will be formally accepted for publication once it meets all outstanding technical requirements.

Kind regards,

Julio Alejandro Henriques Castro da Costa

Academic Editor

PLOS ONE

Additional Editor Comments (optional):

Reviewers' comments:

Reviewer's Responses to Questions

**Comments to the Author**

1. If the authors have adequately addressed your comments raised in a previous round of review and you feel that this manuscript is now acceptable for publication, you may indicate that here to bypass the “Comments to the Author” section, enter your conflict of interest statement in the “Confidential to Editor” section, and submit your "Accept" recommendation.

Reviewer #1: All comments have been addressed

2. Is the manuscript technically sound, and do the data support the conclusions?

Reviewer #1: Yes

3. Has the statistical analysis been performed appropriately and rigorously? 

Reviewer #1: (No Response)

4. Have the authors made all data underlying the findings in their manuscript fully available?

Reviewer #1: (No Response)

5. Is the manuscript presented in an intelligible fashion and written in standard English?

Reviewer #1: Yes

6. Review Comments to the Author

Reviewer #1: Dear Author

I have reviewed the revised version of your manuscript and appreciate the effort you have made to address the reviewers' comment s. The study provides valuable insights into glycemic fluctuations, fatigue, and sleep disturbances in individuals with type 2 diabetes during Ramadan.

However, I believe one critical aspect that could further enhance the comprehensiveness of the study is the evaluation of insulin sensitivity. While the manuscript indirectly addresses this through HbA1c and glycemic fluctuations, a direct measurement of insulin sensitivity (e.g., HOMA-IR or Quicki Index) would have added significant depth to your findings. This would allow a clearer understanding of how factors such as sleep quality, fatigue, and dietary patterns affect insulin resistance and metabolic control.

I strongly recommend that this be included in the limitations section, with a note suggesting future studies to investigate the relationship between insulin sensitivity and the studied variables more comprehensively. For instance:

"One limitation of this study is the lack of direct evaluation of insulin sensitivity, which could provide a more robust understanding of the metabolic mechanisms underlying glycemic fluctuations. Future research should consider including indices such as HOMA-IR or Quicki to analyze the relationship between insulin sensitivity, sleep quality, fatigue, and dietary patterns."

Adding this point would not only acknowledge the scope of the current study but also pave the way for future research to build upon your valuable findings.

Thank you for your contributions to this important field. I look forward to seeing the final version of your work.

Best regards,

7. PLOS authors have the option to publish the peer review history of their article (what does this mean? ). If published, this will include your full peer review and any attached files.

**Do you want your identity to be public for this peer review?** For information about this choice, including consent withdrawal, please see our Privacy Policy .

Reviewer #1: No

---

## [Editor Report · Acceptance letter]

PONE-D-24-43792R1

PLOS ONE

Dear Dr. Romadlon,

I'm pleased to inform you that your manuscript has been deemed suitable for publication in PLOS ONE. Congratulations! Your manuscript is now being handed over to our production team.

Kind regards,

on behalf of

Dr. Julio Alejandro Henriques Castro da Costa

Academic Editor

PLOS ONE